# Targeting Tat–TAR RNA Interaction for HIV-1 Inhibition

**DOI:** 10.3390/v13102004

**Published:** 2021-10-06

**Authors:** Awadh Alanazi, Andrey Ivanov, Namita Kumari, Xionghao Lin, Songping Wang, Dmytro Kovalskyy, Sergei Nekhai

**Affiliations:** 1Department of Microbiology, College of Medicine, Howard University, Washington, DC 20059, USA; aaanazi@ju.edu.sa (A.A.); namita.kumari@howard.edu (N.K.); swang@howard.edu (S.W.); 2Center for Sickle Cell Disease, College of Medicine, Howard University, Washington, DC 20059, USA; andrey.ivanov@howard.edu (A.I.); xionghao.lin@howard.edu (X.L.); 3Department of Oral and Maxillofacial Pathology, College of Dentistry, Howard University, Washington, DC 20059, USA; 4Department of Biochemistry, University of Texas Health Science Center, San Antonio, TX 78229, USA; d.kovalskyy@gmail.com; 5Department of Medicine, College of Medicine, Howard University, Washington, DC 20059, USA

**Keywords:** HIV-1 transcription, HIV-1 Tat, TAR RNA, small molecule inhibitors

## Abstract

The HIV-1 Tat protein interacts with TAR RNA and recruits CDK9/cyclin T1 and other host factors to induce HIV-1 transcription. Thus, Tat–TAR RNA interaction, which is unique for HIV-1, represents an attractive target for anti-HIV-1 therapeutics. To target Tat–TAR RNA interaction, we used a crystal structure of acetylpromazine bound to the bulge of TAR RNA, to dock compounds from the Enamine database containing over two million individual compounds. The docking procedure identified 173 compounds that were further analyzed for the inhibition of HIV-1 infection. The top ten inhibitory compounds with IC_50_ ≤ 6 µM were selected and the three least toxic compounds, T6780107 (IC_50_ = 2.97 μM), T0516-4834 (IC_50_ = 0.2 μM) and T5628834 (IC_50_ = 3.46 μM), were further tested for HIV-1 transcription inhibition. Only the T0516-4834 compound showed selective inhibition of Tat-induced HIV-1 transcription, whereas the T6780107 compound inhibited equally basal and Tat-induced transcription and the T5628834 compound only inhibited basal HIV-1 transcription. The compounds were tested for the inhibition of translation and showed minimal (<25%) effect. The T0516-4834 compound also showed the strongest inhibition of HIV-1 RNA expression and p24 production in CEM T cells and peripheral blood mononuclear cells infected with HIV-1 IIIB. Of the three compounds, only the T0516-4834 compound significantly disrupted Tat–TAR RNA interaction. Additionally, of the three tested compounds, T5628834 and, to a lesser extent, T0516-4834 disrupted Tat–CDK9/cyclin T1 interaction. None of the three compounds showed significant inhibition of the cellular CDK9 and cyclin T1 levels. In silico modelling showed that the T0516-4834 compound interacted with TAR RNA by binding to the bulge formed by U23, U25, C39, G26,C39 and U40 residues. Taken together, our study identified a novel benzoxazole compound that disrupted Tat–TAR RNA interaction and inhibited Tat-induced transcription and HIV-1 infection, suggesting that this compound might serve as a new lead for anti-HIV-1 therapeutics.

## 1. Introduction

Eradication of the human immunodeficiency virus (HIV) remains a challenge despite the introduction of combination anti-retroviral therapy (cART) that suppresses viral load to undetectable levels but does not eliminate viral reservoirs [1]. Interruption of the cART regimen, which is common among HIV-1+ patients, especially in resource-poor countries, results in the HIV-1 rebound and the appearance of a mutated viral species resistant to antiviral drugs [2]. In addition, chronic long-term HIV-1 infection leads to cardiovascular and neurological diseases, metabolic syndrome and non-infectious respiratory disease, which summarily contribute to HIV-1 morbidity and mortality [3,4,5]. As cART does not affect HIV-1 transcription, small molecules are needed to block HIV-1 transcription in order to prevent HIV-1 provirus activation and eliminate HIV-1-related chronic comorbidities. HIV-1 transcription is activated by the HIV-1 Tat protein that interacts with the HIV-1 transactivation response (TAR) RNA and recruits host CDK9/cyclin T1 and other factors to induce the elongation of HIV-1 transcription and facilitate full-length transcript production [6]. In the absence of the Tat protein, HIV-1 transcription is terminated after the synthesis of 82 nucleotide long TAR RNA [7,8]. HIV-1 TAR RNA has been thought as a target for small molecule inhibitors, as it stands out as a unique HIV-1 target. Computation screening of close to 200,000 compounds identified acetylpromazine that was shown to bind the bulge of TAR RNA [9], overlapping with the binding of the Tat protein that specifically interacts with the bulge [10,11]. A more recent study that utilized a virtual screening based on “scaffold-hopping” with several potential TAR RNA binding scaffolds including acetylpromazine, identified molecules that disrupted Tat–TAR RNA interaction with mid–micro molar IC_50_ [12]. Here, we utilized the existing NMR structure of TAR RNA with the bound acetylpromazine to screen in silico compounds from the Enamine database. Docking identified 173 compounds that were further tested for the inhibition of HIV-1 replication in CEM T cells infected with VSVG-pseudotyped HIV-1. This allowed narrowing down of the candidate list to three compounds that were analyzed for their effect on Tat-induced and basal HIV-1 transcription and disruption of the Tat/TAR RNA and Tat/CDK9/cyclin T1 complexes. Taken together, our study identified a novel compound that may serve as a new lead for anti-HIV-1 therapeutics.

## 2. Materials and Methods

### 2.1. Materials

293T and CEM T cells were purchased from ATCC (Manassas, VA). Protein A/G agarose beads, anti-cyclin T1 rabbit monoclonal antibody, anti-CDK9 rabbit monoclonal antibody were purchased from Santa Cruz Biotechnology (Santa Cruz, CA). Anti-FLAG mouse monoclonal antibody was purchased from Sigma (St. Louis, MO, USA). Supplemented DMEM and RPMI media were purchased from Invitrogen. Biotinylated WT TAR RNA (59 nucleotides long) and mutant delta TAR RNA (48 nucleotides long) were synthesized by Integrated DNA Technologies (Coralville, IL, USA). Unless indicated, all other chemicals and enzymes were obtained from Sigma-Aldrich (St. Louis, MO, USA). Small molecules were synthesized by Enamine (Ukraine).

### 2.2. Viruses and Plasmids

HIV-1 proviral DNA, pNL4-3.Luc.R-E containing two nonsense frame shifts in the *env* and *vpr* genes with Luciferase reporter gene cloned in place of *nef* was obtained from NIH AIDS Reagent Program (Germantown, MD). VSVG pseudotyped HIV-1 expressing luciferase virus (HIV-1-LUC-G) was generated using pNL4A3.Luc.R.E and VSVG-expressing plasmid, pHEF-VSVG, obtained from NIH AIDS Reagent Program (Germantown, MD). Flag-Tat-expressing vector was previously described [13]. HIV-1 LTR-luciferase vector was kindly provided by Dr. Manuel López-Cabrera (Unidad de Biología Molecular, Madrid, Spain). 

### 2.3. Molecular Docking

Virtual screening using docking technique of Enamine stock database was conducted with QXP program, which showed excellent results to reproduce experimental docking positions on targets with diverse binding sites. The structure of HIV-1 TAR with bound acetylpromazine (PDB ID: 1LJV, Figure 1A) was used in this study [9]. The binding site was defined and, consequently, grid was generated using residues within 10 Å from the bound ligand. A systematic docking routine sdock+ was employed to generate 300 positions per ligand and top 10 scored positions were saved. The post docking filtering was performed using multiRmsd, which allows performing hypothesis driven selection of putative complexes from QXP docking results.

### 2.4. HIV-1 Infection

HIV-1-LUC-G was utilized to infect leukemic lymphoid CEM T cells. CEM T cells were grown in RPMI medium (Invitrogen) supplemented with 10% Fetal Bovine Serum (FBS) and 1% antibiotic solution (penicillin and streptomycin) at 37 °C in humidified incubator with 5% CO_2_. CEM T cells were infected by HIV-1-LUC-G at a multiplicity of infection (MOI) 0.01, cultured at 1 × 10^6^ cells/mL in 6-well plates at 37 °C and 5% CO_2_ for 24 h and then treated with different concentrations of indicated compounds. For infection with T-tropic HIV-1 IIIB, CEM T cells were infected at a MOI = 0.01. Cells were collected after 6 h post infection for non-integrated HIV-1 DNA isolation and 4 days post infection for p24 and viral RNA analysis.

Peripheral blood mononuclear cells (PBMCs) were activated prior to HIV-1 infection with phytohemagglutinin (PHA) (0.5 μg/mL) for 24 h in complete RMPI media followed by interleukin (IL)-2 (10 U/mL) for 24 h. PBMCs were infected with HIV-1 IIIB (MOI = 0.01). Media and the cells were collected 4 days post infection for p24 measurement and viral RNA analysis.

### 2.5. Luciferase Assay

At 48 h post infection, the cells were collected, washed with PBS and then incubated with 100 μL of steady light reconstituted luciferase buffer (Luclite Kit, PerkinElmer) for 10 min at room temperature. Luminescence was measured using Glo-Max Microplate Multimode reader (Promega).

### 2.6. Cell Proliferation Assays

CEM T cells were seeded at concentration of 2 × 10^5^ cells/mL in 96-well plates and treated with selected inhibitors at different concentrations (1 μM, 3 μM 10 μM, 30 μM, 60 μM and 100 μM) for 24 h. Next day, 10 μL of 3-(4,5-dimethylthiazol-2-yl)-2,5-diphenyltetrazolium bromide (MTT) reagent was added to each well and incubated for 2 h at 37 °C in tissue culture incubator. Then 200 μL of DMSO was added to each well and the absorbance at 595 nm was measured using a microplate reader (Bio-Rad).

Viability of HIV-1 IIIB infected PBMCs was determined using trypan blue exclusion assay. The cells were supplemented with 0.2% trypan blue, transferred to a plastic disposable counting chamber, and counted on a TC10 automatic cell counter (Bio-Rad).

### 2.7. Tat-Induced HIV-1 Transcription Based on Luciferase Assays

293T cells (ATCC) were grown in DMEM media supplemented with 10% Fetal Bovine Serum (FBS) and 1% antibiotic solution (penicillin and streptomycin) at 37 °C in humidified incubator with 5% CO2. 293T cells were seeded in 96-well plates at 30% confluence and Lipofectamine 3000 reagent (Life Technologies) was utilized for transfection. 293T cells were co-transfected with Tat-expressing vector and HIV-1 LTR-luciferase reporter gene. At 24 h post transfection, the compounds were diluted in DMEM media and added to the cells at different concentrations and incubated for another 24 h. At 48 h post transfection, the cells were collected and washed with phosphate-buffered saline (PBS), and then 100 μL of steady light reconstituted luciferase buffer (Lucilyte, PerkinElmer) was added for 10 min at room temperature. Luminescence was measured using Glo-Max Microplate Multimode reader (Promega). 

### 2.8. Tat–TAR RNA Interaction Assay

Biotinylated WT TAR RNA (59 nucleotides long, 5′-GGGUCUCUCUGGUUAGACCAGAUCUGAGCCUGGGAGCUCUCUGGCUAACUAGGGAACCC-3′) and mutant delta TAR RNA with the deletions of Tat-binding bulge nucleotides 21-27 and 38-41 (48 nucleotides long, 5′-GGGUCUCUCUGGUUAGACCAGCCUGGGAGCUGGCUAACUAGGGAACCC-3′) were synthesized by Integrated DNA Technologies (Coralville, IL, USA). To prepare cell lysates containing Tat, 293T cells were co-transfected with Flag-Tat expression vector for 48 h and then lysed in the whole cell lysate buffer (50 mM Tris-HCl, pH 7.5, 0.5 M NaCl, 1% NP-40, 0.1% SDS) supplemented with protease inhibitors cocktail (Sigma). To bind TAR RNA, streptavidin-agarose beads were washed with the binding buffer prepared in RNAse free water (20 mM Tris-HCl pH 7.5, 2.5 mM MgCl_2_, 100 mM NaCl). The beads were blocked with BSA and tRNA was also added to prevent unspecific binding. Blocked beads were then incubated with 10 µg of TAR RNA or 10 µg of delta TAR RNA as control. Beads incubated with TAR RNA and delta TAR RNA were washed and incubated with approximately 400 µg Tat lysate in presence of TAK buffer (50 mM Tris-HCl, pH 8.0, 5 mM MgCl_2_, 5 mM MnCl_2_, 10 μM ZnSO_4_, 1 mM DTT). Then 10 µM of each compound was added. Beads were collected by 5 min centrifugation at 1000× *g*, and the bound proteins were eluted in 1x SDS loading buffer. Proteins then separated on 10% Bis-Tris gel, followed by transferring to PVDF membrane, immunoblotting with anti-Flag antibody and chemiluminescence detection as previously described [14].

### 2.9. Immunoprecipitation and Western Blotting Analysis

293T cells transfected as described above were lysed in the whole cell lysate buffer supplemented with protease inhibitors as described above. Total protein concentration was determined using BCA protein assay (Bio-Rad). Lysates were stored at -80 ℃ until use. Protein A/G agarose beads (Sana Cruz Biotechnology) were blocked with 5% Bovine Serum Albumin (BSA). Blocked beads were incubated with anti-Flag antibody then combined with protein lysate and incubated for 2 h at 4 °C. Beads were washed with 1x TNN buffer (10 mM Tris-HCl 7.5, 0.1% NP-40 and 100 mM NaCl) and then heated for 5 min at 95 °C. Eluted proteins were loaded on 10% Bis-Tris gel and resolved in 1x MOPS running buffer supplemented with SDS and anti-oxidant (Invitrogen). Proteins in the gel were transferred to PVDF membrane. The membrane was blocked in 5% non-fat milk, washed in PBS buffer containing 0.1% Tween-20 (PBST) and incubated with primary antibodies including Anti-CDK9 (1:4000) (Rabbit, Lot No: I0414), Anti-Cyclin T1 (1:4000) (Rabbit, Lot No:D1709) or anti-Flag 1:2000 for 2 h at room temperature. The membrane was washed with PBST and incubated with HRP-conjugated anti-mouse or anti-rabbit and probed. Chemiluminescence was detected with chemiluminescent substrate (Clarity ECL Western Blot Substrate Kits, Bio-Rad) using ChemiDoc XRS Imaging station (Bio-Rad).

### 2.10. The p24 Enzyme-Linked Immunosorbent Assay (ELISA)

CEM T cells or PBMCs were infected with HIV-1 IIIB (MOI = 0.01). Supernatants were collected at 48 h post infection and p24 was measured by HIV-1 p24 antibody ELISA (PerkinElmer) using OPD as substrate. Data were analyzed using standards provided within the kit. 

### 2.11. Determination of HIV-1 gag and nef mRNA and HIV-1 Reverse Transcription

Quantitative analysis of HIV-1 RNA was conducted on total RNA isolated from CEM T cells infected with fully replication-competent HIV-1(IIIB). RNA was isolated using TRIzol Reagent (Invitrogen, Carlsbad, CA). For *gag* or *nef* RNA quantification, total RNA (100 ng) was reverse transcribed to cDNA using Superscript RT-PCR kit (Invitrogen, Carlsbad, CA). For quantification of HIV-1 DNA, total DNA was extracted from 1 × 10^6^ cells infected with HIV-1(IIIB) using total DNA isolation kit (Thermo Fischer). Real-time PCR analysis was conducted in Roche Light Cycler 480 (Roche Diagnostics) using SYBR Green Master Mix (Roche Diagnostics). PCR amplification was carried with initial preincubation for 5 min at 45 ℃, then 3 min at 95 ℃ followed by 45 cycles of denaturation at 95 ℃ for 15 s, annealing and extension at 60 ℃ for 45 s, and final extension at 72 ℃ for 10 s. Mean Cp values for Early LTR, Late LTR transcripts and β-globin DNA was determined using ΔΔ*C*t method. Quantification of gag or nef mRNA was carried out using 18S RNA as a housekeeping control. Mean *C*p values were determined and ΔΔ*C*t method was used to calculate relative expression levels. The following primer sequences were used.

HIV gag forward: ATAATCCACCTATCCCAGTAGGAGAAAT

HIV gag reverse: TITGGTCCITGTCITATGTCCAGAATGC

HIV nef forward: ATCCACTGACCTTTGGATGG

HIV nef reverse: GTACTCCGGATGCAGCTCTC

Early LTR forward: GGCTAACTAGGGAACCCACTG

Early LTR reverse: CTGCTAGAGATTTTCCACACTGAC

Late LTR forward: TGTGTGCCCGTCTGTTGTGT

Late LTR reverse: GAGTCCTGCGTCGAGAGATC

β-actin forward: CTCCCAAAGTGCTGGGATTA

β-actin reverse: CAAAGGCGAGGCTCTGTG

18SrRNA forward: CAAAGGCGAGGCTCTGTG

18SrRNA reverse: CTCCAGGTTTTGCAACCAGT

### 2.12. In Vitro Translation System

Rabbit reticulocytes lysate translation system (Promega) was used to test the effect of compounds on translation. The 50 µL reaction contained 35 µL reticylocyte lysate (10 mM creatine phosphate, 50μg/mL creatine phosphokinase, 2 mM DTT, 50 μg/mL calf liver tRNA, 79 mM Potassium Acetate, 0.5 mM Magnesium acetate, 0.02 mM Hemin), 2 µM amino acids, Ribonuclease inhibitor (1.1 U/μL), 43 ng luciferase RNA and 3 µg Transcend™ tRNA. The indicated inhibitors were added at 30 µM concentration and incubated at 30 ℃ for 90 min. The reaction was terminated by placing the reaction on ice. Biotin containing translation products were analyzed by Western blotting as described above using the antibodies provided in the kit.

### 2.13. Statistical Analysis

Statistical analysis was performed using GraphPad Prism 6 software (Graph Pad Software, San Diego, CA, USA). All data are presented as means with standard deviation. Means and standard deviations were calculated for 3 repeats at each concentration of each compound. Differences between the two groups were compared using the parametric unpaired two-tailed Student’s t-test. For reproducibility, experiments were repeated three times. 

## 3. Results

### 3.1. Selection of TAR RNA Binding Compounds by Molecular Docking

In this study, we aimed at discovering small molecules that would form extensive hydrophobic contacts and pi–pi stacking interactions with TAR RNA. We chose the existing TAR RNA complex with acetylpromazine [9] as acetylpromazine was proposed to bind TAR RNA with high affinity (nanomolar) intercalating into the RNA and forming both polar and extensive hydrophobic and pi–pi interactions (Figure 1A). Additionally, acetylpromazine was predicted to bind the bulge of TAR RNA, which would allow it to compete with Tat binding. The full Enamine stock database (over 2 million compounds) was pre-filtered to extract structures containing one hydrogen bond donor and one aromatic ring. This initial filtering yielded 500,136 compounds. Molecular docking results were processed to extract complexes that matched the following criteria: (i) a putative ligand should intercalate into RNA structure (ii) form a hydrogen bond with a phosphate oxygen of U25 and (iii) form stacking interactions with bases of C26, C39, U40 and U25. The filtering was performed using docking processing suite multiRmsd software that allows performing hypothesis-driven selections of putative complexes from QXP docking results. As a result, we selected 173 compounds that were screened in vitro.

### 3.2. Screening Small Molecules Library for Inhibition of One round of HIV-1 Infection

To evaluate the HIV-1 inhibitory activity of TAR RNA-targeting small molecules, CEM T cells were infected with VSVG pseudotyped pNL4-3.Luc.R-E-virus (HIV-1-LUC-G) for 24 h and then treated with 10 μM compounds for another 24 h. Screening of 173 compounds showed nine compounds that inhibited HIV-1 below 40% of control (Figure 1B and Appendix A). 

### 3.3. Analysis of Cytotoxicity and Dose-Dependent Inhibition of One round of HIV-1

Next, we determined IC_50_s for one round of HIV-1 inhibition of the nine identified compounds. CEM T cells were infected with HIV-1-LUC-G for 24 h followed by treatment with the selected compounds for another 48 h (Figure 2A,B). The three compounds that showed consistent HIV-1 inhibition were T0516-4834 (IC_50_ = 0.3 μM), T6780107 (IC_50_ = 3.7 μM) and T5628834 (IC_50_ = 3.7 μM) (Figure 2A). These compounds inhibited HIV-1 with lower IC_50_s compared to the previously reported HIV-1 transcription inhibitor, 1E7-03 (IC_50_ = 8.7 μM) [15] (Figure 2A). The rest of the compounds showed either upregulation of HIV-1 infection at lower concentrations or inefficient inhibition (Figure 2B). None of the compounds showed any toxicity under 100 μM concentrations when tested in 3-(4,5-dimethylthiazol-2-yl)-2,5-diphenyltetrazolium bromide (MTT) assay (Figure 2C).

### 3.4. Effect of the Selected Compounds on Tat-Mediated Transcription and Production of HIV-1 gag mRNA

The three selected compounds (Figure 3A) were further analyzed for their mechanism of action. To test whether the compounds interfered with basal or HIV-1 Tat activated transcription, we utilized a luciferase reporter assay in 293T cells that were transfected with HIV-1 LTR-luciferase reporter plasmid alone or in combination with Flag-tagged HIV-1 Tat-expressing plasmid. At 24 h post transfection, the cells were treated with different concentrations of the compounds for another 24 h. 

Basal HIV-1 transcription was inhibited in a dose-dependent manner by the T6780107 compound (IC_50_ = 4.63 μM) and T5628834 compound (IC_50_ = 6.2 μM), whereas the T0516-4834 compound had little effect (IC_50_ = 15 μM) (Figure 3B). Co-expression of HIV-1 LTR reporter and Flag-Tat-expressing plasmid resulted in >40-fold induction of transcription by Tat (Appendix A). Dose-dependent inhibition of Tat-induced transcription was observed for the T0516-4834 compound (IC_50_ = 2 μM) and T6780107 compound (IC_50_ = 3.5 μM), whereas the T5628834 compound showed little inhibition (IC_50_ = 20 μM) (Figure 3C). Thus, the T0516-4834 compound selectively inhibited Tat-induced transcription; the T6780107 compound affected equally both basal and Tat-induced transcription and the T5628834 compound showed only inhibition of basal but not Tat-induced transcription. 

To elucidate whether the selected compounds have an effect on HIV-1 during infection with fully replication-capable HIV-1, CEM T cells were infected with T-tropic HIV-1 IIIB and p24 levels and *gag* mRNA expression was quantified by ELISA and real-time RT-PCR (Figure 3D,E). The strongest reduction in p24 levels (4-fold reduction) and HIV-1 gag mRNA (27-fold reduction) was observed for the T0516-4834 compound (Figure 3D,E). The T6780107 compound showed 2.2-fold p24 reduction and 5.5-fold reduction of *gag* mRNA (Figure 3D,E). The T5628834 compound showed 1.26-fold reduction in p24 level and 1.6-fold reduction in gag mRNA with no statistical significance (Figure 3D,E).

To determine whether the effect of compounds could be due to the inhibition of translation, we analyzed their effect in vitro using a non-radioactive reticulocyte translation system with tRNA^Lys^ pre-charged with ε-labeled biotinylated lysine. The addition of 30 μM cycloheximide completely blocked translation (Appendix A). Among the tested compounds, only 30 μM of the T0516-4834 compound showed a statistically significant(~30%) reduction in luciferase translation. However, this inhibitory effect was not sufficient to explain the strong (>90%) inhibition of HIV-1 transcription and replication. 

To further characterize the antiviral activity of the selected compounds in primary cells, we tested their effect on PBMCs infected with HIV-1 IIIB. We observed the strongest reduction in p24 (4-fold reduction) and *gag* mRNA (2.5-fold reduction) in the T0516-4834 treated PBMCs (Figure 3F,G). No significant effect was observed for T6780107, whereas T5628834 treatment showed only p24 reduction (3.2-fold reduction) (Figure 3F,G). We also observed a strong (10-fold) reduction in HIV-1 *nef* mRNA by T0516-4834 (Appendix A).

We also analyzed the effect of the three compounds on the viability of PBMCs infected with HIV-1 IIIB. Trypan blue exclusion assay showed that none of the tested compounds had any significant effect on cell viability in contrast to azidothymidine (AZT) and raltegravir that not only strongly inhibited HIV-1 (Figure 3G) but also reduced PBMCs viability (AZT, 29% reduction and raltegravir, 18% reduction) (Figure 3H).

Taken together, these observations suggested that the T0516-4834 compound was the best overall HIV-1 inhibitor with a specific effect on Tat-induced HIV-1 transcription and no toxicity.

### 3.5. Effect of the Selected Compounds on HIV-1 Reverse Transcription

We next analyzed the effect of the compounds on the early steps of HIV-1 infection. CEM T cells were infected with HIV-1 IIIB. HIV-1 DNA was isolated at 6 h post infection and quantified using primers for early and late RT products. AZT used as positive control reduced HIV-1 early LTR products as expected (Appendix A, 1.8-fold reduction) and late LTR products (Appendix A, 63-fold reduction). No significant reduction was observed for either of the tested compounds compared to AZT (Appendix A).

### 3.6. Disruption of Tat–TAR Interaction by T0516-4834

As we were targeting TAR RNA, we next tested whether the selected compounds had an effect on the interaction between Tat and TAR RNA. Biotinylated 59 nt TAR RNA was immobilized on avidin-containing agarose beads and used to pull down Tat from lysates of 293T cells that were transfected with Flag-Tat-expressing vector (Figure 4A). As a negative control, we used 48 nt TAR RNA with the deletion of bulge (delta TAR RNA, Figure 4A). We observed Tat binding to the immobilized TAR RNA (Figure 4B, lane 2) but not the beads alone (Figure 4B, lane 3) or the beads-immobilized delta TAR RNA (Figure 4B, lane 4). We observed significant disruption of Tat binding to TAR RNA with the addition of 10 µM of the T0516-4834 compound (2.5-fold inhibition, *p* = 0.001) (Figure 4B, lane 8 and Figure 4C). In contrast, two other compounds showed little effect on Tat–TAR RNA binding (Figure 4B,C). 

### 3.7. Effect of the Compounds on Tat/CDK9/Cyclin T1 Complex Formation

As only one compound disrupted Tat–TAR RNA interaction, we extended our analysis to evaluate whether other inhibitory compounds might have an effect on Tat interaction with CDK9/cyclin T1 complex and, thus, inhibit HIV-1 transcription. We utilized immunoprecipitation assays using cell lysates collected from 293T cells co-transfected with Flag-Tat for 24 h, followed by treatment with the 20 μM compounds for another 24 h. Immunoprecipitations were carried out with protein A/G agarose beads that were pre-blocked with BSA and then incubated with cell lysates. Anti-Flag antibody was used to precipitate Flag-tagged Tat and the Tat-associated proteins that were eluted from the beads with SDS-PAGE loading buffer resolved on SDS-PAGE and detected with different antibodies. Tat protein expression levels were slightly increased in the cells treated with any of the tested compounds (Figure 5A). Among the tested compounds, T5628834 had the strongest effect (4-fold reduction) on CDK9 associated with Tat (Figure 5B). Compound T0516-4834 showed a moderate reduction (1.4-fold) in CDK9 associated with Tat and compound T6780107 had no effect (Figure 5B). Cyclin T1 protein levels associated with Tat were reduced by the T5628834 compound (1.6-fold reduction), whereas the T0516-4834 compound showed less than 20% reduction and the T6780107 compound had no significant effect (Figure 5C). To exclude the effect of the compounds on the cellular levels of CDK9 and cyclin T1, we additionally analyzed CDK9 and cyclin T1 in the cells treated with the compounds. We found no biologically significant reduction in the levels of CDK9 (Appendix A) or cyclin T1 (Appendix A).

### 3.8. Model of TAR RNA Binding for T0516-4834

To visualize the T0516-4834 compound interaction with TAR RNA, we built an in silico model that shows binding of the discovered hit to the HIV-1 TAR RNA bulge formed by U23, U25, C39, G26,C39 and U40 (Figure 6A). The top performing compound, benzoxazole T0516-4834, binds into the crevice formed by these nucleotides forming a pi–pi staking interaction with U25 and U40 and its chlorine atom forms van der Waals contacts with the cleft formed by C39 and U40 (Figure 6A). The T0516-4834 compound showed similar binding as compared to a thienopyridine compound 2 [16], which interacts with TAR RNA within the same binding cavity and whose butterfly shape of the heterocycle allows formation of extensive pi–pi stacking interactions with the U25, G26 and U40 bases (Figure 6B).

Taken together, we identified a novel molecule that prevented Tat interaction with TAR RNA and efficiently inhibited Tat-induced HIV-1 transcription and viral replication. 

## 4. Discussion

In this study, we screened over one million individual compounds from the Enamine database against the TAR RNA structure, and identified 173 compounds that were further analyzed to identify the most efficient inhibitor and to gain insight on its mechanism of action. Our top candidate, the T0516-4834 compound showed selective inhibition of Tat-induced HIV-1 transcription, efficient inhibition of HIV-1 *gag* mRNA expression and p24 production and disruption of Tat–TAR RNA interaction and, to a lesser extent, Tat/CDK9/cyclin T1 interaction. While the T0516-4834 compound inhibited both unspliced *gag* mRNA as well as spliced *nef* mRNA production, the T6780107 compound only inhibited unspliced *gag* mRNA production, suggesting that it may affect HIV-1 splicing. A recent study has reported the discovery and optimization of thienopyrimidines as HIV-1 TAR RNA binding molecules [16]. These TAR RNA binding molecules displaced Tat-derived peptide from TAR RNA with 40 μM IC_50_ and demonstrated non-canonical TAR RNA binding as determined by NMR [16]. Our top performing compound benzoxazole T0516-4834 showed in silico interaction with TAR RNA within a cleft formed by U25 and U40 nucleotides, similar to the thienopyridine compound 2 [16] (Figure 6). Thienopyridines inhibited the HIV-1-induced cytopathic effect at 28 µM EC_50_, without cytotoxicity [17]. In our study, we observed HIV-1 inhibition with 0.3 μM IC_50_ in one round of HIV-1 infection. We also achieved ~10-fold HIV-1 inhibition in HIV-1 IIIB infected PBMCs treated with 10 µM T0516-4834. Thus our compound described here seems to show superior HIV-1 inhibition despite its close structure similarity to the previously described compound 2. In our preliminary comparison of the T0516-4834 compound to the thienopyridine compound 2, we conducted a series of 100 ns MD simulations, which revealed that compound 2 preserves pi–pi interaction with G26 and C39 only, while bases from the opposite side of the crevice (bases U25 and U40) divert from the ligand and their interactions are transient. In turn, T0516-4834 reshapes the crevice and forms stable pi–pi stacking with bases G26, C39 and U40 and these interactions are preserved through the entire simulation. Thus, T0516-4834 compound seems to induce RNA rearrangement (induced fit) that might allow maintaining more extensive interactions with the TAR RNA molecule. These data will be published elsewhere.

In addition to thienopyrimidines, several additional classes of molecules were shown to bind TAR RNA including a cyclic peptide mimic of Tat [18], amilorides [19] and a tri-cationic oligopyridylamide, ADH-19 [20]. However, none of these compounds were able to achieve significant HIV-1 inhibition with sub micromolar or even low micromolar IC_50_. 

To understand whether the identified compound selectively inhibited Tat-induced transcription, we analyzed its effect on basal and Tat-induced transcription. Only the T6780107 and T5628834 compounds affected HIV-1 basal transcription with low micro molar IC_50_s. Tat-induced transcription was inhibited by T6780107 and T0516-4834, whereas the T5628834 compound showed little effect. Thus, only the T0516-4834 compound showed selectivity for the inhibition of Tat-mediated transcription compared to basal transcription. These effects are consistent with the inhibition of Tat–TAR RNA interaction by the T0516-4834 compound, which only happens at Tat-induced and not basal transcription. The T5628834 compound suppressed p24 levels and *gag* mRNA in HIV-1 infected PBMCs but had no effect on *nef* mRNA production. Thus, while T5628834 might disrupt binding of Tat to CDK9/Cyclin T1, it might have additional effects, such as an effect on HIV-1 splicing. Moreover, the T6780107 compound might have an effect on splicing as it reduced p24 and *gag* mRNA levels but had no effect on *nef* mRNA. This compound had no effect on CDK9/cyclin T1 interaction and remains to be further investigated.

The previously reported TR87 compound, which binds with high affinity to TAR RNA, reduces Tat-mediated transcription and downregulates HIV-1 replication through the disturbance of Tat/TAR RNA interaction [21]. Similarly, CGP64222, which binds Tat protein inducing a conformational change in TAR RNA bulge, blocks HIV-1 replication and specifically inhibits Tat-transactivation [22,23]. The most promising, to date, HIV-1 transcription inhibitor, didehydro-cortistatin A, targets Tat protein by binding to the positively charged lysine of Tat’s TAR RNA binding domain and decreases the binding affinity of Tat to TAR RNA [24]. 

The tested compounds inhibited HIV-1 transcription and had no effect on HIV-1 reverse transcription. Thus, the early steps of HIV-1 infection including entry, binding and reverse transcription are not likely to be affected by the compounds. The utility of HIV-1 transcription inhibitors is highlighted by our recent study in which the activation of HIV-1 in macrophages was the major contributing factor to lung inflammation in HIV-1 transgenic (HIV-Tg) mice [25,26]. Treatment of HIV-Tg mice with 1E7-03, an HIV-1 transcription inhibitor, ameliorated LPS-induced lung inflammation and prevented mice death [26]. As individuals living with HIV-1 have a higher prevalence of non-infectious lung disease [27], future testing of the compounds described here are warranted in HIV-1-related pathologies including lung inflammation. Because cART therapy has no effect on HIV transcription, the development of transcriptional inhibitors such as T0516-4834 or 1E7-03 is needed and might help to treat HIV-1–related chronic complications.

## 5. Conclusions

Our present study successfully identified the T0516-4834 compound that downregulated HIV-1 transcription and replication and disrupted Tat binding to TAR RNA. Moreover, T0516-4834 was found to be selective for Tat-induced transcription. Our study confirms that disruption of Tat–TAR RNA interaction is a feasible approach for the inhibition of HIV-1 transcription and replication. The identified compound might serve as a new lead for anti-HIV-1 inhibitory molecules. 

## Figures and Tables

**Figure 1 viruses-13-02004-f001:**
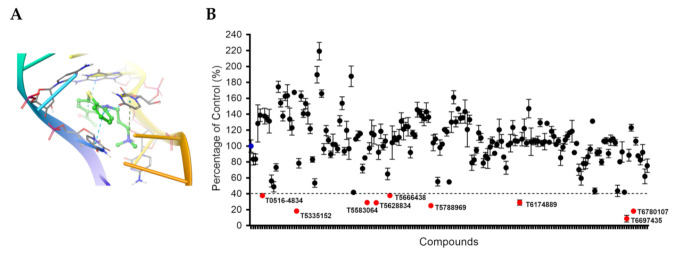
Screening of TAR RNA-targeting compounds. (**A**) Complex of acetylpromazine with TAR RNA (PDB ID: 1LVJ). (**B**) High-throughput screening for one round of HIV-1 infection. HIV-1-LUC-G infected CEM T cells were seeded in 96-well plates and treated with 10 μM concentration of compounds. DMSO represents 100% (blue) and percentage of inhibition was calculated for each compound. In red, compounds are shown that achieved more than 60% inhibition.

**Figure 2 viruses-13-02004-f002:**
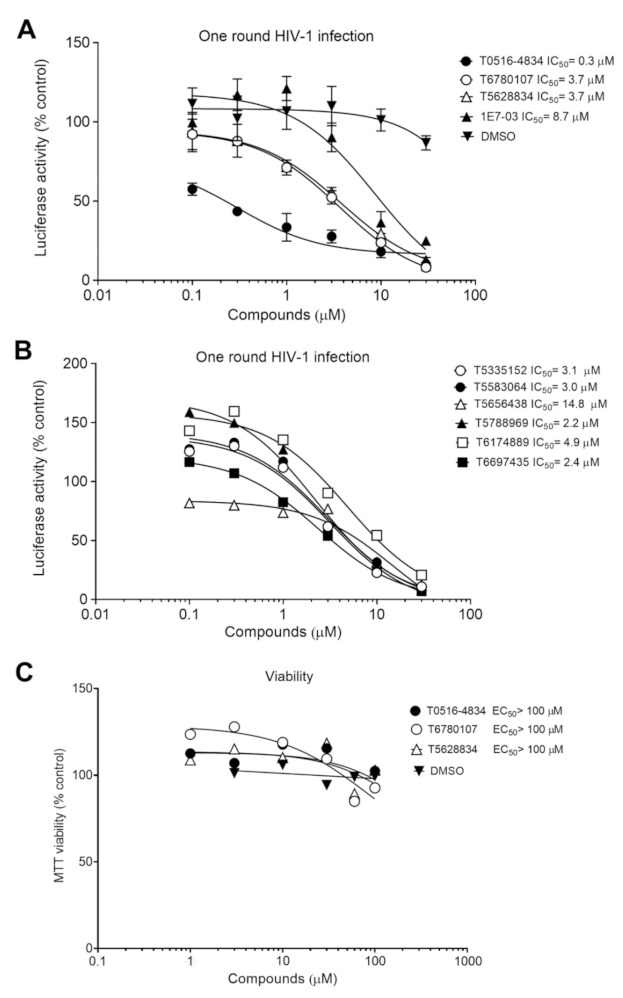
Dose-dependent HIV-1 inhibition by top candidate compounds. (**A**,**B**) CEM T cells were infected with HIV-1-LUC-G, then seeded in 96-well plate and treated with the indicated concentrations of the compounds. DMSO represents 100% and percentage of inhibition was calculated for each compound concentration. (**C**) CEM T cells were seeded in 96-well plates and treated with the indicated concentrations of the compounds. MTT was added and the formazan absorbance measure as described in Materials and Methods. DMSO represents 100% and percentage of viability was calculated for each compound concentration.

**Figure 3 viruses-13-02004-f003:**
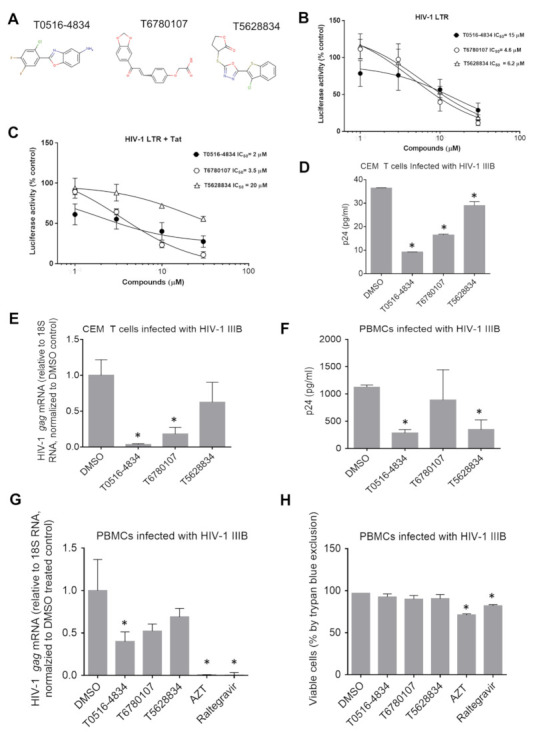
Inhibition of Tat-medicated transcription and HIV-1 replication by TAR RNA-targeting compounds. (**A**) shown are structures for T0516-4834, T6780107 and T5628834 compounds. (**B**,**C**) inhibition of HIV-1 basal and Tat-activated transcription. 293T cells were transfected with HIV-1 LTR reporter (**B**) or HIV-1 LTR reporter and Tat-expressing vectors (**C**) and treated with indicated concentrations of the compounds for 24 h. Luciferase activity was measured as described in Materials and Methods. Data were plotted in Prims 6. (**D**,**E**) inhibition of HIV-1 replication in CEM T cells infected with HIV-1 IIIB. The cells were infected for 48 h followed by p24 ELISA (**D**) or RNA isolation and real-time PCR analysis for HIV-1 *gag* mRNA (**E**). Asterisks indicate *p* < 0.05 in comparison to the DMSO controls. F and G, inhibition of HIV-1 replication in PBMCs infected with HIV-1 IIIB. PBMCs were activated as described in Materials and Methods and infected with HIV-1 IIIB for 48 h followed by p24 ELISA analysis (**F**) or RNA isolation and real-time PCR analysis for HIV-1 *gag* mRNA (**G**). (**H**) PBMCs viability tested with trypan blue exclusion assay for samples from (**G**). Asterisks indicate *p* < 0.05 in comparison to the DMSO controls.

**Figure 4 viruses-13-02004-f004:**
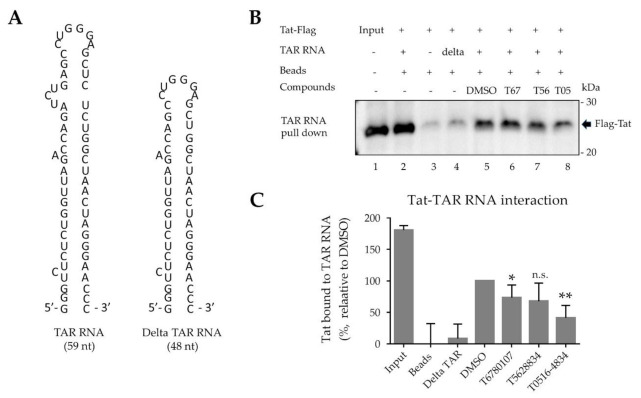
Disruption of Tat–TAR RNA interaction by T0516-4834 compound. (**A**) Structures of TAR RNA and bulge-deleted (delta) TAR RNA. (**B**,**C**) Immunoblotting of Tat bound to TAR RNA. Tat and TAR RNA binding was performed using protein lysate prepared from 293T cells expressing Flag-Tat and biotinylated TAR RNA coupled to agarose beads. Where indicated, the incubation mixture was supplemented with 10 µM of the indicated compound. Tat protein was eluted from the beads with SDS loading buffer, resolved on SDS-PAGE and detected with anti-Flag antibody. (**C**) Quantification of Tat protein bound to TAR RNA. Shown are results from three independent experiments that were plotted using Prism 6. * *p* = 0.04 and ** *p* = 0.001 in comparison to the DMSO controls.

**Figure 5 viruses-13-02004-f005:**
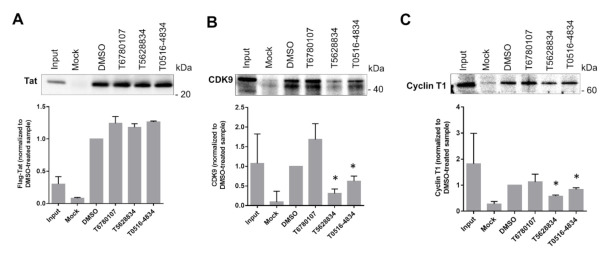
Effect of TAR RNA-targeting compounds on Tat interaction with CDK9/Cyclin T1. Tat precipitated from 293T cells co-transfected with Flag-Tat and, where indicated, treated with 20 μM compounds. Protein levels were determined by Western blotting. (**A**) Tat expression determined by anti-Flag antibodies. (**B**) CDK9 protein expression determined using anti-CDK9 antibodies. (**C**) Level of cyclin T1 determined by anti-cyclin T1 antibodies. Quantification was performed using Prism 6 from three independent experiments. Asterisks indicate *p* < 0.01 in comparison to the DMSO controls.

**Figure 6 viruses-13-02004-f006:**
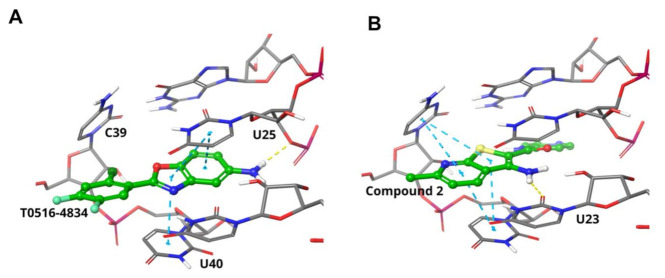
Binding models of HIV-1 TAR RNA complexed with T0516-4834 (**A**) and previously reported thienopyridine compound 2 [16] (**B**). Nucleotides that form the binding cavity are shown in sticks. Pi–pi stacking is shown as light blue dashed lines.

## Data Availability

Information on the compounds structure can be obtained upon request from the authors.

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
