# Peer review of "Targeting Tat–TAR RNA Interaction for HIV-1 Inhibition"

_viruses, 2021, doi:10.3390/v13102004_

Round 1

Reviewer 1 Report

The revised manuscript by Alanzi et al has addressed most of the issues raised in the previous review. They provide new data assessing the toxicity of the compounds in multiple contexts and the ability of the compounds to inhibit HIV-1 replication in primary cells. Of note is the striking difference in performance of the compounds in the various assays systems used. While T0516-4834 displays the greatest activity in all cell based assays, there are marked differences in the performance of T5628834 versus T6780107. While T6780107 shows greater anti-HIV-1 activity in CEM T cells, it has no activity in PBMCs. In contrast, T5628834 has limited activity in CEM T cells but reduces p24 expression in PBMCs (to a similar level as T0516-4834) but does not alter HIV-1 nef RNA expression, suggesting a different mode of action (see Fig. 3). Subsequent in vitro experiments attempt to evaluate the basis for the responses observed. In Fig. 4, Tat-TAR RNA pulldowns indicate that neither T5628834 nor T6780107 affected Tat binding while T0516-4834 induced only a modest reduction (~2 fold). Parallel tests of the effect of compounds on the interaction between Tat and cyclin T1/cdk9 determined that T5628834 resulted in the greatest reduction in co-immunoprecipitated cyclin T1/Cdk9 (~4 fold) followed by T0516-4834 (~2 fold). T6780107 either had no effect or slightly increased binding. Subsequent modeling (Fig. 6) looked at the possible interaction of T0516-4834 with TAR-RNA but no analysis is provided of the interaction of the related compound T5628834 to explore why it has such limited activity.

Overall, the authors have provided clear evidence that T0516-4834 is a promising inhibitor of HIV-1 replication acting through the disruption of Tat transactivation. One caveat is the marked differences in the performance of T5628834 versus T6780107 in the various assays and cell lines. Given such cell specific differences, can one extrapolate results from HEK293 cells (i,e. the pulldown assays) to results in CEM T cells and PBMCs?

Minor corrections

Fig. 3E. Vertical axis should indicate that it is HIV-1 gag RNA that is being measured.

Legend of Fig. 3 needs to be corrected as 3G is referred to as representing HIV-1 p24 and nef RNA.

Unclear why RT-qPCR was used to measure gag RNA in fig. 3E and nef RNA was measured in fig. 3G, particularly given that the protein measured to monitor HIV-1 expression is Gag. Given the discordance between the effects on viral protein and RNA levels upon T5628834 addition, it might have been better to perform analysis on the various viral RNAs (unspliced, singly spliced, and multiply spliced). The effect of T5628834 may be due to modulation of unspliced viral RNA levels that would not have be detected using a nef primer set.

Fig. 5 Analysis would benefit from measurement of the effect of compound treatment on total levels of cyclin T1 and Cdk9 in cell lysates. Effects on levels of cyclin T1 or Cdk9 in the pulldown assay could be due to altered expression of these proteins.

Fig. 6. Structural modelling of T0516-4834 and compound 2 with TAR RNA is shown but there is limited discussion of why one is more active than the other  in the biological assays.

Author Response

Reviewer 1

Comments and Suggestions for Authors:  

The revised manuscript by Alanazi et al has addressed most of the issues raised in the previous review. They provide new data assessing the toxicity of the compounds in multiple contexts and the ability of the compounds to inhibit HIV-1 replication in primary cells. Of note is the striking difference in performance of the compounds in the various assays systems used. While T0516-4834 displays the greatest activity in all cell based assays, there are marked differences in the performance of T5628834 versus T6780107. While T6780107 shows greater anti-HIV-1 activity in CEM T cells, it has no activity in PBMCs. In contrast, T5628834 has limited activity in CEM T cells but reduces p24 expression in PBMCs (to a similar level as T0516-4834) but does not alter HIV-1 nef RNA expression, suggesting a different mode of action (see Fig. 3). Subsequent in vitro experiments attempt to evaluate the basis for the responses observed. In Fig. 4, Tat-TAR RNA pulldowns indicate that neither T5628834 nor T6780107 affected Tat binding while T0516-4834 induced only a modest reduction (~2 fold). Parallel tests of the effect of compounds on the interaction between Tat and cyclin T1/cdk9 determined that T5628834 resulted in the greatest reduction in co-immunoprecipitated cyclin T1/Cdk9 (~4 fold) followed by T0516-4834 (~2 fold). T6780107 either had no effect or slightly increased binding. Subsequent modeling (Fig. 6) looked at the possible interaction of T0516-4834 with TAR-RNA but no analysis is provided of the interaction of the related compound T5628834 to explore why it has such limited activity.

Overall, the authors have provided clear evidence that T0516-4834 is a promising inhibitor of HIV-1 replication acting through the disruption of Tat transactivation. One caveat is the marked differences in the performance of T5628834 versus T6780107 in the various assays and cell lines. Given such cell specific differences, can one extrapolate results from HEK293 cells (i,e. the pulldown assays) to results in CEM T cells and PBMCs?

Response: We thank the reviewer for the overall positive assessment of our study. We agree that we did not explore further the related compound T5628834. Most likely it will require some additional structural experiments, such as NMR of TAR RNA with the bound compound. We will keep this in mind for future exploration. We agree that it is hard to extrapolate the data on 293T cells to cultured and primary T cells. However, we argue that only T0516-4834 compound showed selective inhibition of Tat-induced HIV-1 transcription in 293T cells (>5-fold deceases in IC50 versus basal transcription). The other two compounds had no selectivity (T6780107,  less than 2 fold difference) or no inhibition of Tat-induced transcription (T562834). Also, only T0516-4834 compound disrupted Tat-TAR RNA interaction in lysates obtained from 293T cells expressing Tat.  Thus we are confident in our main conclusion that T0516-4834 compound is the best HIV-1 inhibitor and its mechanism includes the disruption of Tat – TAR RNA interaction.

Minor corrections

Comment 1. Fig. 3E. Vertical axis should indicate that it is HIV-1 gag RNA that is being measured.

Response: We thank the reviewer for noticing this omission. The y axis legends were corrected for Fig. 3E and Fig. 3G to indicate that mRNA is being measured.

Comment 2. Legend of Fig. 3 needs to be corrected as 3G is referred to as representing HIV-1 p24 and nef RNA.

Response: we corrected the legend (also swapped nef for gag, see our response below).

Comment 3. Unclear why RT-qPCR was used to measure gag RNA in fig. 3E and nef RNA was measured in fig. 3G, particularly given that the protein measured to monitor HIV-1 expression is Gag. Given the discordance between the effects on viral protein and RNA levels upon T5628834 addition, it might have been better to perform analysis on the various viral RNAs (unspliced, singly spliced, and multiply spliced). The effect of T5628834 may be due to modulation of unspliced viral RNA levels that would not have be detected using a nef primer set.

Response: We agree and substituted panel G for HIV-1 gag mRNA analysis to be consistent with the p24 data. We moved nef mRNA results to the supplemental data and added discussion about the possibility that T5628834 compound might have an effect on HIV-1 splicing rather than transcription. We believe that analysis of RNA splicing is beyond the scope of this study. However, we are thankful for this suggestion and will keep this in mind for future analysis.

Comment 4. Fig. 5 Analysis would benefit from measurement of the effect of compound treatment on total levels of cyclin T1 and Cdk9 in cell lysates. Effects on levels of cyclin T1 or Cdk9 in the pulldown assay could be due to altered expression of these proteins.

Response: we agree and added this analysis that showed no biologically significant reduction of CDK9 or cyclin T1 levels (supplemental Figure 5).

Comment 5. Fig. 6. Structural modelling of T0516-4834 and compound 2 with TAR RNA is shown but there is limited discussion of why one is more active than the other in the biological assays.

Response: We agree and added discussion: “. In our preliminary comparison of T0516-4834 compound to the thienopyridine compound 2, we conducted a series of 100 ns MD simulations which revealed that compound 2 preserves pi-pi interaction with G26 and C39 only, while bases from the opposite side of the crevice (bases U25 and U40) divert from the ligand and their interactions are transient. In turn, T0516-4834 reshapes the crevice and forms stable pi-pi stacking with bases G26, C39 and U40 and these interactions are preserved through the entire simulation. Thus  T0516-4834 compound seems to induce RNA rearrangement (induced fit) what might allow it to maintain more extensive interactions with the TAR RNA molecule.  These data will be published elsewhere.”

Reviewer 2 Report

In this manuscript, Alanazi et al., searched for compounds from the Enamine database that target the interaction that exists between the HIV-1 TAR RNA element and the viral TAT protein in order to inhibit viral transcription and HIV-1 replication. Three of them were selected after screening and analyzed for their ability to inhibit viral replication and transcription and to disrupt the interaction between TAR/TAT as well as TAT/CDK9/Cyclin. The work is very interesting and well conducted but it requires additional experiments and modifications to be published. Some parts of the manuscript are too descriptive and require more discussion and analysis such as data from fig5.

Main comments:

The section 3.1 seems to rely on the data published by Du et al 2002 chemistry & biology (ref #4). This need to be clarified and I recommend the author to include a figure.

Section 3.2: please provide additional information for Fig1 as a table (supplementary data) which indicates the reference of all tested compounds and their corresponding the luciferase assay result. some could be of interest for the community. Some points do not have error bars. How many replicates have been performed? (same question for fig.2 and 3)

Section 3.4: Fig.3B and 3C do not show an inhibition of the transcription step but an inhibition of the luciferase activity that could be due to a decrease of the translational rate. RT-qPCR have to be performed on these extracts for such conclusions. Please provide a control in which the luciferase mRNA is not under control of the LTR promotor and could be not affected by the drug. The differences obtained between Fig.3B and 3C are not highlighted and discussed: why the addition of TAT protein induced a strong diminution of the IC50 for T0516-4834 compounds but had opposite effect for T5628834?

Section 3.5: the authors should show that the interaction between TAT and TAR is disrupted by T05 compound from infected cellular extracts.

section 3.7. I do not understand what was performed by the authors. Did they isolate non-integrated or integrated HIV-1 DNA? Was the experiment performed at 6 or 48hpi? This part needs to be clarified and developed.

Specific comments:

Introduction: Please cite those two articles in addition to ref #7: The two sides of Tat. Barboric M, Fujinaga K. Elife. 2016 Jan 19;5:e12686. doi: 10.7554/eLife.12686; and Ott M, Geyer M, Zhou Q (2011) The control of HIV transcription: keeping RNA polymerase II on track Cell Host & Microbe 10:426–435

Section 2.1: some sentences could be pulled in one: eg “293T and CEM T cells were purchased…”

Section 2.2: cite the reference instead of “[PUBMED ID 12079782]”. What does “PDB ID 1LJV” mean?

Section 2.4: the MOI used for the HIV-1 LUC G strain is missing. The first sentence can be removed to avoid a repetition.

Section 2.6: please define “MTT”

Section 2.9: please rewrite the sentence “the gel was blotted to PVDF membrane and overnight”; define PBST (just 3 lines above).

Section 2.10: the MOI is missing

Section 2.11: the RT step description is missing.

Section 3.3: 1E7-03 inhibitor requires a reference.

Fig.2, Fig.3b and fig.3c: please provide a x axis in µM instead of M, it would be easier for the reader to check the IC50 which is expressed in µM

Section 3.5: Fig4B, lane 2: the upper panel indicates that the TAR RNA is absent while the text indicates the opposite, and there is an “input” instead of a “cross”. Two independent experiments are not sufficient for statistical analysis.

Section 3.7: the term "component 2" is too vague and imprecise, please specify pyridinyl analogues as indicated in ref#10.

Author Response

Reviewer 2.

Comments and Suggestions for Authors

In this manuscript, Alanazi et al., searched for compounds from the Enamine database that target the interaction that exists between the HIV-1 TAR RNA element and the viral TAT protein in order to inhibit viral transcription and HIV-1 replication. Three of them were selected after screening and analyzed for their ability to inhibit viral replication and transcription and to disrupt the interaction between TAR/TAT as well as TAT/CDK9/Cyclin. The work is very interesting and well conducted but it requires additional experiments and modifications to be published. Some parts of the manuscript are too descriptive and require more discussion and analysis such as data from fig5.

Response: We thank the reviewer for the overall positive assessment of our study. We responded in details to the comments below.

Main comments:

Comment 1. The section 3.1 seems to rely on the data published by Du et al 2002 chemistry & biology (ref #4). This need to be clarified and I recommend the author to include a figure.

Response: We agree and added a figure panel (revised Figure1A).

Comment 2. Section 3.2: please provide additional information for Fig1 as a table (supplementary data) which indicates the reference of all tested compounds and their corresponding the luciferase assay result. some could be of interest for the community. Some points do not have error bars. How many replicates have been performed? (same question for fig.2 and 3)

Response: We agree and added a Supplementary Table 1 showing corresponding data plotted on Figure 1B. All measurements were done in triplicates (this also applied to Figs 2 and 3).

Comment 3. Section 3.4: Fig.3B and 3C do not show an inhibition of the transcription step but an inhibition of the luciferase activity that could be due to a decrease of the translational rate. RT-qPCR have to be performed on these extracts for such conclusions. Please provide a control in which the luciferase mRNA is not under control of the LTR promotor and could be not affected by the drug. The differences obtained between Fig.3B and 3C are not highlighted and discussed: why the addition of TAT protein induced a strong diminution of the IC50 for T0516-4834 compounds but had opposite effect for T0516-4834 compound

Response: we agree and tested the effect of the compounds on translation of luciferase mRNA in reticulocyte lysates (supplemental Figure 2). We did not observe any biologically significant effect. We include this in the discussion. We also added  discussion about the effect of the compounds on basal and Tat-induced transcription: “To understand whether the identified compound selectively inhibited Tat-induced transcription, we analyzed its effect on basal and Tat-induced transcription. Only T6780107 and T5628834 compounds affected HIV-1 basal transcription with low micro molar IC50s. Tat- induced transcription was inhibited by T6780107 and T0516-4834, whereas T5628834 compound showed little effect. Thus, only T0516-4834 compound showed selectivity for the inhibition of Tat-mediated transcription compared to basal transcription.  These effects are consistent with the inhibition of Tat-TAR RNA interaction by T0516-4834 compound, which only happens at Tat-induced and not the basal transcription. The T5628834 compound suppressed p24 levels and gag mRNA in HIV-1 infected PBMCs but had no effect on nef mRNA production. Thus, while T5628834 might disrupt binding of Tat to CDK9/Cyclin T1, it might have additional effects such as an effect on HIV-1 splicing. Also, T6780107 compound might have an effect on splicing as it reduced p24 and gag mRNA levels, but had no effect on nef mRNA. This compound had no effect on CDK9/cyclin T1 interaction and remains to be further investigated.”

 Comment 4. Section 3.5: the authors should show that the interaction between TAT and TAR is disrupted by T05 compound from infected cellular extracts.

Response: we respectfully disagree. It is notoriously difficult to detect Tat in the infected lysates. Moreover, we intentionally used Flag-expressing Tat to be sure that we can detect Tat as anti-Tat antibodies are, in general, of poor quality. As the lysate is being used only to supply Tat, we do not see how this additional in vitro experiment will add to the existing story.

Comment 4.  Section 3.7. I do not understand what was performed by the authors. Did they isolate non-integrated or integrated HIV-1 DNA? Was the experiment performed at 6 or 48hpi? This part needs to be clarified and developed.

Response: we isolated HIV-1 non-integrated DNA at 6 hrs post infection. We corrected the Methods to indicate the correct time (6 hrs). 

Specific comments:

Comment 5.  Introduction: Please cite those two articles in addition to ref #7: The two sides of Tat. Barboric M, Fujinaga K. Elife. 2016 Jan 19;5:e12686. doi: 10.7554/eLife.12686; and Ott M, Geyer M, Zhou Q (2011) The control of HIV transcription: keeping RNA polymerase II on track Cell Host & Microbe 10:426–435

Response: we thank the reviewer for noticing this. We found and fixed several misplaced references and added the requested references in place of Reference 7.

Comment 6. Section 2.1: some sentences could be pulled in one: eg “293T and CEM T cells were purchased…”

Response: corrected

Comment 7.  Section 2.2: cite the reference instead of “[PUBMED ID 12079782]”. What does “PDB ID 1LJV” mean?

Response: we added citation and changed “PDB ID 1LJV” to “PDB ID: 1LJV” to indicate the PDB ID of TAR RNA-promazine structure.

Comment 8.  Section 2.4: the MOI used for the HIV-1 LUC G strain is missing. The first sentence can be removed to avoid a repetition.

Response: MOI is added.

Comment 9. Section 2.6: please define “MTT”

Response: defined.

Comment 10. Section 2.9: please rewrite the sentence “the gel was blotted to PVDF membrane and overnight”; define PBST (just 3 lines above).

Response: we corrected the sentence and defined PBST.

Comment 11. Section 2.10: the MOI is missing

Response: MOI is added.

Comment 12. Section 2.11: the RT step description is missing.

Response: this information is at the end of this section, we moved it to the front to be clearer.

Comment 13. Section 3.3: 1E7-03 inhibitor requires a reference.

Response: the reference is added.

Comment 14. Fig.2, Fig.3b and fig.3c: please provide a x axis in µM instead of M, it would be easier for the reader to check the IC50 which is expressed in µM

Response: X scales converted to µM.

Section 3.5: Fig4B, lane 2: the upper panel indicates that the TAR RNA is absent while the text indicates the opposite, and there is an “input” instead of a “cross”. Two independent experiments are not sufficient for statistical analysis.

Response: the figure legend was corrected. We also added an additional experiment for statistical quantification.

Section 3.7: the term "component 2" is too vague and imprecise, please specify pyridinyl analogues as indicated in ref#10.

Response: we agree and changed “compound 2” to “thienopyridine compound 2”, based on the description of this compound in the reference.

Round 2

Reviewer 2 Report

In this revised manuscript, Alanazi and colleagues have provided many additional elements (new data and modifications) that make the article robust and clear for the reader. However, the analysis of the figure related to section 3.4 remains inaccurate. As I mentioned in my previous report, the results of fig.3B and 3C show only a decrease of the luciferase activity in the presence of the different component. The authors cannot directly conclude from these data that the “basal HIV-1 transcription was inhibited in dose-dependent manner” by the different components. They should modify the text according the result (second paragraph of the section). The following figures (Fig.3C to 3H and suppl. figures) are consistent with an inhibition of the HIV-1 transcription.

Specific comments:

Section 2.8: typing error, “SDS” instead of “SDSO”

Author Response

Specific comments:

Section 2.8: typing error, “SDS” instead of “SDSO”

Response: we fixed the error. Thank you!